# Developing of SiO_2_ Nanoshells Loaded with Fluticasone Propionate for Potential Nasal Drug Delivery: Determination of Pro-Inflammatory Cytokines through mRNA Expression

**DOI:** 10.3390/jfb13040229

**Published:** 2022-11-08

**Authors:** Yasir Mehmood, Hira Shahid, Md Abdur Rashid, Yahya Alhamhoom, Mohsin Kazi

**Affiliations:** 1Department of Pharmaceutics, Faculty of Pharmaceutical Sciences, Government College University Faisalabad, Faisalabad P.O. Box 38000, Pakistan; 2Department of Pharmacology, Faculty of Pharmaceutical Sciences, Government College University Faisalabad, Faisalabad P.O. Box 38000, Pakistan; 3Department of Pharmaceutics, College of Pharmacy, King Khalid University, Guraiger, Abha 62529, Saudi Arabia; 4Pharmacy Discipline, Faculty of Health, School of Clinical Sciences, Queensland University of Technology, Brisbane, QLD 4000, Australia; 5Department of Pharmaceutics, College of Pharmacy, King Saud University, Riyadh 11451, Saudi Arabia

**Keywords:** nasal spray, in-vitro, cell viability, controlled, crystalline

## Abstract

Mesoporous Silica Nanoparticles (MSN) are porous inorganic materials that have been extensively used for drug delivery due to their special qualities, such as biocompatibility, biodegradability, and non-toxicity. MSN is a promising drug delivery system to enhance the efficacy and safety of drug administration in nasal diseases like chronic rhinitis (CR). In this study, we used the sol-gel technique for MSN synthesis and incorporate fluticasone propionate (FP) for intranasal drug administration for the treatment of chronic rhinitis (CR). In order to confirm the particle size, shape, drug release, and compatibility, various instruments were used. MSN was effectively prepared with average sizes ranging between 400 ± 34 nm (mean ± SD) as measured by dynamic light scattering (DLS), while zeta potential verified in all cases their positive charged surface. To investigate MSN features, the Fourier transform infrared spectrometer (FTIR), scanning electron microscopy (SEM), transmission electron microscope (TEM), thermal analysis, X-ray diffraction (XRD), and nitrogen adsorption/desorption measurement were used. The loaded compound was submitted to in vitro dissolution tests, and a remarkable dissolution rate improvement was observed compared to the crystalline drug in both pH conditions (1.2 and 7.4 pH). By using an MTT assay cell viability was assessed. The expression levels of the anti-inflammatory cytokines IL-4 and IL-5 were also measured using mRNA extraction from rat blood. Other characterizations like acute toxicity and hemolytic activity were also performed to confirm loaded MSN safety. Loaded MSN was incorporated in nasal spray prepared by using innovator excipients including poloxamer. After this, its nasal spray’s physical characteristics were also determined and compared with a commercial product (Ticovate).

## 1. Introduction

Chronic rhinitis (CR) is a common inflammatory disease of the nasal passages and sinuses that can significantly impair a person’s daily function and quality of life [1,2]. It is generally characterized by nasal congestion and runny nose lasting at least 12 weeks [2]. Treatment approaches for patients with CR involve a complex combination of surgical and pharmacological therapies, but topical and systemic steroid use is fundamental to managing this condition [3]. Endoscopic sinus surgery (ESS) treats CR, refractory to conventional medical therapy, by promoting sinus airway patency and clearing the sinuses while reducing the severity of inflammation. It is widely recognized as the surgical procedure of choice [4]. Even while surgery isn’t usually straightforward and painless, it can have risks like significant postoperative bleeding, mucosal adhesions, infection, inflammation, and subpar surgical results. Contrarily, because of their powerful anti-inflammatory characteristics steroids are frequently utilized in the treatment of CR [5]. However, while oral steroids are effective, their use is associated with several adverse effects [6]. Due to their favorable safety profile, topical steroids are being used more frequently, and both patients and doctors prefer intranasal delivery using a variety of nasal sprays [7]. Intranasal steroids are considered to be the cornerstone of CR therapy because they provide high local drug concentrations with minimal systemic exposure [8,9]. It has been shown that fluticasone propionate (FP) works well to treat nasal inflammatory diseases [10]. It has been utilized for sinus reasons in clinical practice for more than 20 years, and a wealth of literature demonstrates its therapeutic characteristics for such illnesses [11]. FP is a highly lipophilic, sparingly water-soluble (0.0108 mg/mL) drug with a relatively low risk of systemic absorption [12]. Bioavailability is reported to be less than 11%, but a study of repeated doses of inhaled corticosteroids suggests a bioavailability of 11% [12]. The 0.1% ointment has a bioavailability of 0.7%. The terminal half-life of an inhaled dose is approximately 5 h, while other sources report 5.8 h. The poor water solubility of drugs is one of the biggest problems in drug development [13]. Insoluble drugs require excipients to enhance solubilities, such as surfactants, co-solvents, micellar solutions, chelating agents, and lipid formulations [14,15]. Unfortunately, there is a need to resolve the issues of insolubility more effectively to reduce the side effects of molecules [16]. 

Certain anatomical, physiological, chemical, and therapeutic obstacles associated with conventional dosage forms can be bypassed by nanotechnology-based drug administration [14,17]. Various key therapeutic agents, including nucleic acids, peptides, and tiny hydrophobic or hydrophilic molecules, can be delivered into different biological systems by means of nanoparticles [18]. Potential advantages of NPs include enhanced drug solubility and stability, enhanced bioavailability at target sites, and extended duration of action through controlled release rates [19]. These properties enhanced drug loading and release capabilities. To have biomedical applications, such particles must have specific particle sizes, pore sizes, shapes, and pore geometries [20]. Nanotechnology-based drug delivery can overcome certain anatomy [17]. There are numerous types of nanoparticles [21]. Among these nanoparticles, silica nanoparticles possess more attraction due to their good properties [22,23]. The use of silica as a substrate in nasal drug delivery and pulmonary drug delivery has a special advantage. Due to it being biocompatible, silica is frequently employed in inorganic nanoparticles. Silica nanoparticles are very porous and highly tuneable in terms of pore size, density, and total surface area, they are great tools for creating drug delivery systems [24]. This system has several advantages over polymeric nano delivery systems like its huge drug loading, less leakage of drugs, olfactory mucosal, and lung biocompatible [25]. It is also stable in olfactory mucosal pH (5.5–6.5), which can increase rhinitis (7.2–8.2) [26]. In several studies, silica nanoparticles were used for pulmonary drug delivery because of their good inhaled properties, and after administration, silica nanoparticles can reach the target tissue or target site (like tumors, lungs, etc.). After reaching the target site it can be engulfed by the mononuclear phagocyte system where it is degraded by the hydrolyzation and silica matrix changed into orthosilicic acid which returns to blood circulation and finally is removed from the body [25].

Silica can be synthesized by various preparation techniques in the form of NPs, transparent films, or solid flat materials [27]. Silanols and siloxanes group which are present on the surface of silica particles can be triggered by synthesis modification [28]. The aim of this work was to formulate a poorly soluble, short-half-life drug-loaded MSN intended for intranasal administration in order to control release and increase drug loading capacity.

## 2. Materials and Methods

### 2.1. Chemicals

Tetraethyl orthosilicate (TEOS), acetonitrile, hydrochloric acid, and Cetyltrimethylam-monium bromide (CTAB) were purchased from Dae-Jung chemicals (Seohaean-ro Sinan, Gyeonggi, Korea). Fluticasone Propionate was graciously provided by Saffron Pharmaceuticals (Faisalabad, Pakistan). Spectrum Medical Industries sold dialysis membranes having a molecular weight cutoff (MWCO) of 8000 Da (Houston, TX, USA). Pharmaceutical-grade materials poloxamer, Microcrystalline, Cellulose (MCC), and Sodium Carboxymethylcellulose (CMC) were employed (JRS PHARMA GmbH & Co. KG Holzmühle, Rosenberg, Germany). Throughout the studies, double-distilled and purified water that was manufactured on-site was used. Ticovate nasal spray (Saffron Pharma, Faisalabad, Pakistan) was purchased from the market andwas used for comparison. The rest of the reagents and solvents were all analytical-grade and utilized exactly as they were given to us.

### 2.2. Preparation of Mesoporous Silica Nanoparticles

We recently reported using a modified sol-gel technique to synthesize mesoporous silica nanoparticles [29]. We selected MSN produced with a 1:1 acetonitrile to water ratio for this study from many trials because of their ideal size and form. In a 250 mL flask, 100 mL of an acetonitrile-water (1:1) combination was stirred for 15 min at 35 °C. After that, 300 mg of CTAB was continuously poured into the aforementioned solution while maintaining the same temperature and nitrogen atmosphere. After creating a clear solution, 5 mL TOES was gradually added using a 10 mL syringe while being constantly stirred for an hour. The reaction started as soon as the resulting solution swiftly turned opaque or milky white. The white gel (MSN) was filtered using a 0.1-micron Sartorius filter while compressing with nitrogen. It was then washed three times with deionized water and allowed to air dry for eight hours at room temperature. The dried bulk was then entirely stripped of the surfactant template by being calcined at 500 °C for 6 h.

### 2.3. Drug Loading in MSN

Weighted fluticasone propionate was dissolved in methanol at a concentration of 100 mg/100 mL. FP solution and 100 mg of MSN were combined while being stirred at 1500 rpm for three hours and then stored for three days. The loading and storing were done in closed containers outside in the open air. To fully remove the methanol, the mixture was then dried at 55 °C for more than a day. The dried composite samples, referred to as MSN-FP, were washed with water and again dried. Drug loading sketch was shown in Figure 1.

### 2.4. Preparation of Nasal Spray Nano Formulation by Using Fluticasone Propionate Loaded MSN

Nasal spray formulations were prepared using MSN preloaded with fluticasone propionate (MSN-FP). This formulation used the same excipients as the original product, microcrystalline cellulose (MCC) and sodium Carboxymethylcellulose (CMC). Poloxamer was also used. Purified water was placed in a large container, then CMC and MCC were added to the container, and homogenization was started. After 2 h, the poloxamer was added to the mixture and continuously mixed and homogenized. After 1 h, slowly add fluticasone propionate (MSN-FP) while continuing to stir. Once a uniform mix is achieved, begin filling the container with the nasal spray. 

### 2.5. Determination of Entrapment Efficiency

Each 100 mg of MSN-FP was measured with an analytical balance. MSN-FP was separately combined with 100 mL of methanol while being gently stirred for an hour. A validated HPLC method was used to quantify the filtrate after the combination had been filtered. The Shimadzu LC 20AB HPLC system had a UV-VIS detector and a 1000-pressure pump (Shimadzu Scientific Instruments, Kyoto, Japan). Ammonium acetate buffer (pH = 7.6), methanol, and acetonitrile were employed as the mobile phase in a ratio of 10:50:40 (%*v*/*v*) on an Agela C18 column (250 × 4.6 mm, 5 µ) as the stationary phase. The UV detector was set at a wavelength of 254 nm, the injection volume was 20 µL, and the mobile phase flow rate was 1.0 mL/min. Drug loading efficiencies were computed in triplicate and compared with the resulting chromatograms to standards. The nasal spray stability and FP leakage from MSN within the nasal spray were both determined using the same process.
(1)$$\mathrm{Entrapment}\;\mathrm{efficiency}\;(\%)=\frac{Drug\;added-Free\;drug}{Drug\;added}\times 100$$

### 2.6. Particle Size, Polydispersity Index (PDI) and Zeta Potentials

Using a particle size analyzer and photon correlation spectroscopy (PCS), the average particle size, PDI, and zeta potential were calculated (Zetasizer, Malvern Instruments, Malvern, UK). In this technique, nano nasal sprays were made in filtered water and used right away to assess the PDI and particle size.

### 2.7. FTIR Spectroscopy

In this study, we used an FTIR spectrometer from the USA (Thermo Scientific Nicolet iN5 FTIR) equipped with an attenuated total reflection crystal cell. In this method, a spatula is used to place a small amount of sample on the instrument and clamp the sample. There was no need to prepare a sample for this FTIR. Using this method, spectra of MSN, FP, poloxamer, MSN-FP, and nano nasal spray were recorded in the range 400–4000 cm^−1^.

### 2.8. DSC and TGA Study

Pure drugs, MSN and MSN-FP, were subjected to scanning calorimetry using a Diamond Series DSC from PerkinElmer (Waltham, Massachusetts, USA). The sample was placed in a standard empty aluminum crucible with a hole drilled in the center and then scanned over a temperature range from 30 to 300 °C at a rate of 10 °C min^−1^ under a nitrogen atmosphere. TGA was also performed on a pure drug, MSN and MSN-FP. 

### 2.9. XRPD Study

The pure drug, MSN and MSN-FP were exposed to an X-ray powder diffraction analyzer made by D8ADVANCE, Bruker, Germany. The radiation was Cobalt filtered Fe^2+^ having a wavelength of 1.7890°A. Data were recorded from 2θ angle of 5° to 50° at a step size of 0.02° and scanning speed of 4°/min.

### 2.10. Morphology STUDY

Morphological analysis was performed using a scanning electron microscope (JEOL Ltd., Tokyo, Japan). The pure drug, MSN-FP, and MSN-FP within nasal spray were examined through SEM using adhesive tape coated with gold using a sputter coater. At 200 kV TEM images were taken with a JEOL JEM 2100F field Emission Gun. Almost for 15 min, the MSN-FP sample was washed in (Elma S 30 H) ultrasonic bath with ethanol before the assay. 

### 2.11. Nitrogen Adsorption–Desorption Analysis

Utilizing a Gemini VII 2390 surface area analyzer (Micromeritics Instrument Corp., Norcross, GA, USA) operating at −196.15 °C, the micro-meritic parameters, including surface area, pore volume, and size, were measured. Prior to analysis, the material was degassed at 200 °C for 24 h. Adsorption-desorption data were subjected to the Barrett-Joyner-Halenda (BJH) and Brunauer-Emmett-Teller (BET) techniques to evaluate pore properties [30].

### 2.12. In-Vitro Release Study

The dialysis membrane method was used to conduct in vitro release tests on the medicinal material and nano nasal spray. Briefly, a dialysis bag (MWCO 12-14000 Daltons, Medicell International Ltd., 239 Liverpool Road, (Liverpool, UK) was filled with a nasal spray containing 1 mg of medication and was sealed in a pouch. The dialysis bags were then individually suspended in 500 mL of media maintained at 37 ± 0.5 °C with continuous agitation at 50 rpm, containing 0.1 M HCl, pH 1.2, and phosphate buffer, pH 7.4. At specified intervals up to 8 h, samples of 2.0 mL were obtained, and the same volume of fresh medium was added while maintaining the same conditions. The resulting filtrate was collected from a 0.45 µm pore size, diluted with mobile phase, and used for HPLC analysis. The concentration of the drug in formulated samples was quantified using a calibration curve (R^2^ = 0.998) generated during method validation studies [31]. The percentage release of the drug was calculated by following Equation (2) [32,33]:(2)$$\mathrm{Drug}\;\mathrm{release}\;\mathrm{percentage}=\frac{\mathrm{sample}\;\mathrm{absorbance}}{\mathrm{standard}\;\mathrm{absorbance}}\times 100$$

### 2.13. In-Vitro Diffusion Studies

This experiment was performed by using Franz Diffusion Cell. Silicon membrane was used in this experiment. The permeation through the silicon membrane of the MSN-FP and the pure drug was evaluated in vitro, using a vertical Franz diffusion cell (SES Analytical Systems, GmbH, Germany), with a donor surface area of 1.2 cm^2^ and a receptor volume of 5.2 mL. The semisynthetic membrane (silicon SAMCO (Nuneaton, UK) was placed between the donor and receptor chambers of the Franz diffusion cell having the epidermal side up. The donor chamber of 1.0 mL volume covered with paraffin was filled with a sample. While the receptor chamber has phosphate buffer saline of pH 7.4 at 37 ± 0.5 °C temperature circulating water. The membrane was placed in contact with the receptor phase for the time of 0.5 h at a temperature of 37 ± 0.5 °C earlier to the experiment. Samples (0.5 mL) were withdrawn at pre-determined time intervals over 8 h (0.5, 1, 2, 4, 6, and 8 h) of receptor phase and analyzed through a UV spectrophotometer at 254 nm while the cell was re-filled with an equivalent concentration of newly prepared buffer solution [34,35].

### 2.14. In Vitro Cell Viability Studies

The University of Lahore got a human hepatocyte cell line (HepG2) from the American Type Culture Collection (ATCC). At room temperature, 95% (*v*/*v*) humidity, and 5% (*v*/*v*) CO_2_, cells were maintained in DMEM supplemented with 10% (*v*/*v*) FBS. One day before MSN incubation, cells were sown close to confluency. As a measure of cell viability and proliferation, the MTT assay (3-(4,5-dimethylthiazol-2-yl)-2,5-diphenyltetrazolium bromide) measures the activity of the mitochondrial succinate dehydrogenase enzyme [36]. For 24 h, different MSN-FP nano nasal spray doses (50–400 g/mL) were applied to the cell culture medium. Each well was then filled with 15 µL of MTT (5 mg/mL) in phosphate buffered saline (PBS), which was added, and incubated at 37 °C for an additional 4 h. A microplate reader was used to measure the formazan crystals’ absorbance at 570 nm after the medium had been gently aspirated with an MTT, solubilized in 100 µL of dimethylsulfoxide (DMSO), and formed in each well [37]. The percentage of cell viability was determined using the following formula.
$$\%\;cell\;Viability=\frac{Abs\;of\;treated\;cell-Abs\;of\;blank}{Abs\;of\;Control-Abs\;of\;Blank}\times 100$$

### 2.15. Hemolytic Investigations

To conduct the hemolytic study, blood was collected in an ethylene diamine tetra acetic acid-containing tube, centrifuged at 1500 rpm for 5 min, the supernatant was removed and the precipitate was washed 3 times with phosphate buffer saline (PBS). Then, 200 µL of washed blood sediment was added to 3.8 mL of phosphate buffer saline and vortexed for a few minutes. After that, samples were kept at 37 °C for 2 h followed by centrifugation for 5 min at 1600 rpm. Note the supernatant absorbance at 541 nm. In this experiment, Triton-X was selected as the positive control and the phosphate buffer saline was as a negative control, and % hemolysis was determined by using Equation (3) (Assadi et al., 2018).
% *Hemolysis* = *ABS* − *ABS*0/*ABS*100 − *ABS*0 ∗ 100% (3)

### 2.16. Acute Toxicity Studies

According to Organization for Economic Co-operation & Development Toxicity guideline, the animal model was executed for MSN-FP formulation toxicity. Experiments were conducted with ethics approval from the Institutional Research Ethics Committee (Rashid Lateef College of Pharmacy IRB No (RLCP-EP/5/2021) 12 rats weighing 150–170 g were purchased from Tollinton market, Lahore, Pakistan and kept in the animal house of RLCP for fourteen days. Four groups were divided, and three rats were in each group (Control, Diseased, Pure drug, and MSN-FP). Rhinitis was established in rats using an ovalbumin (OVA) sensitization method [38]. Pure drug and MSN-FP were topically administered by nasal route in the test group and survived on water and food in to control group. On the 14th day of the study blood samples were collected and examined various biochemical parameters [39]. 

### 2.17. Determination of Pro-Inflammatory Cytokines IL-4, IL-5 mRNA Expression

Airway inflammation is mediated by chemokines and cytokines. Cytokines are produced by eosinophils, lymphocytes, and mast cells. Pro-inflammatory cytokines (IL-4 and IL-5) produced primarily by Th-2 lymphocytes cause intense inflammation in allergic asthma. These proteins recruit Th-2 lymphocytes, neutrophils, mast cells, and eosinophils. Both eosinophils and mast cells play important roles in the pathogenesis of asthma. These cells produce cytokines and leukotrienes, causing bronchoconstriction. 12 rats weighted 150–170 g were purchased from Tollinton market, Lahore, Pakistan, and kept in the animal house of RLCP. Four groups were divided, and three rats were in each group (Control, Diseased, Pure drug, and MSN-FP). Rhinitis was established in rats using an ovalbumin (OVA) sensitization method [38]. Pure drug and MSN-FP were topically administered by nasal route in the test group and the control group survived on water and food. Tissue mRNA expression levels of the proinflammatory cytokines IL-4 and IL-5 were measured by reverse transcription-polymerase chain reaction. Total RNA was extracted from the lung tissues of rats using the standard TRIzol method. First stand cDNA synthesis was done by reverse transcription, and different components were used to complete it. The resulting prepared cDNA was stored at −20 °C for further use as a template for PCR. The cDNA prepared by reverse transcription was further amplified by polymerase chain reaction. The PCR protocol included denaturation at 95 °C for 10 s, annealing at 58–60 Co for 20 s (35 cycles), and extension at 72 °C for 30 s.

### 2.18. Physical Appearance, Viscosity and pH Determination

MSN-FP nano nasal spray was checked for its physical properties. There on, 1% dispersion prepared in an aqueous medium was tested for pH by using a digital pH meter (Xylem Analytics, GmbH & Co. KG, WTW-Weilheim, Germany). The formulated MSN-FP nano nasal spray was evaluated for its viscosities at 0.5, 1, and 2 rpm by the using Brookfield viscometer (Model DV-II) using Spindle #21.

## 3. Results

In the medical treatment of CR, a chronic inflammatory illness defined by the buildup of very viscoelastic mucus in the sinuses, topical steroids is the first line of treatment. The inadequate distribution of topical steroids to the nose and sinuses, however, frequently results in a restricted therapeutic response. Utilizing MSN for localized medication administration may help ensure constant delivery to desired locations, increase contact time, elevate local drug concentrations, and reduce systemic side effects while improving therapeutic outcomes. Utilizing a modified sol-gel process, MSN-FP was created. In order to improve solubility, MSN-FP was added to the formulation for nasal spray and further investigated for stability.

### 3.1. Particle Size, Polydispersity Index (PDI) and Zeta Potentials 

The size and size distribution of this formulation was determined and characterized using DLS. Figure 2 shows a size distribution diagram of MSN-FP within the water. The DLS result showed that the hydrodynamic diameter of the MSN-FP was 400 nm with PDI of 0.382 and charge was −10 mV.

### 3.2. FT-IR Analysis

Using FTIR spectroscopy in the 400–4000 cm^−1^ spectrum range, the produced MSN were examined for identification and characterization of functional groups contained in silica particles, as shown in Figure 3. Mesoporous silica nanoparticles had characteristic silicate absorption bands at 797, 1053, 1636, and 3396 cm^−1^, which are often attributed to siloxane bonds (797.97) [40], Si-O-Si bending (1053.78 cm^−1^) and silanol (Si-OH) symmetric stretching (3396.49) [41,42] and bending vibrations at 1636 cm^−1^ [40,42]. FP has a distinct FT-IR spectrum as shown in Figure 3. The monohydrate form has a broad and weak peak at 3500–3600 cm^−1^. This is believed to be an O–H stretch of water molecules within the crystal lattice. These characteristic bands reflect the ‘strongly bound’ water or water of crystallization observed in numerous hydrates [43]. This is confirmed by the monohydrate crystal structure. The water molecule is tightly bound to the FP molecule by three types of hydrogen bonds. The three peaks in the carbonyl region between 1650 and 1750 cm^−1^ reflect the three carbonyl groups of the FP molecule. Both forms peak at 1732 and 1658 cm^−1^. However, the 1723 cm^−1^ carbonyl peak of the monohydrate shifts to 1706 cm^−1^. Assigned to ester carbonyls that do not participate in hydrogen bonding [43]. The stretching of C=O was noticed at 1635 cm^−1^ which represents the presence of amide-I while at 1540 cm^−1^ of the amide-II (N-H bend) group in FP. The existence of the carboxylic group was confirmed by a minor peak noted at 1350 cm^−1^. These findings are supported by previous studies [44,45,46].

FTIR of MSN-FP shows a small peak indicating successful loading of FP onto MSN particles. Bending of Si-O-Si (1053.78 cm^−1^) and symmetric stretching of silanol (Si-OH) (3396.49) [41,42] and bending vibrations at 1636 cm^−1^ [40,42] is present in MS-FP and MSN-FP Nano nasal spray, which clearly indicated MSN-FP present into stable form within nasal spray suspension and the drug is present in amorphous form. The vibrational starching at 2950 cm^−1^ belongs to the methanetriyl group. In Figure 3, major peaks were observed at 2900 cm^−1^ and 1710 cm^−1^, which shows an asymmetric stretch FP. In the FTIR of MSN-FP, the alkenyl stretch (C=C) shifted from 1632 cm^−1^ to 1655 cm^−1^. 

### 3.3. Thermal Analysis 

A differential Calorimeter modeled Q-2000 made by TA USA was used to evaluate FP and MSN-FP. The temperature range was 25 °C to 500 °C, adjusted at a rate of heating 20 °C/min. The response of FP and MSN-FP formulation against heat flow was shown in Figure 4. As shown in Figure 4, the DSC curve of FP exhibits an endothermic peak at 50 °C, which corresponds to its intrinsic melting points due to dehydration and elimination of other volatile components. An exothermic peak at about 150 °C is observed, which can be attributed to glass transition temperature T_g_. At approximately 250 and 415 °C, two overlapping endothermic peaks have been found. This thermal process suggested that FP significantly degrades during the melting phase, and it was observed that 8% substantial weight loss was found over the same temperature range (TGA Figure 4). The complex thermal events around the melting point indicate that FP undergoes significant degradation during melting. It agrees with the observed significant weight loss of 8% detected over the same temperature range by TGA Figure 4 [43]. In MSN-FP, no sharp exothermic peak was found at 50 °C, but an exothermic peak was found at the same position and temperature that recognized that FP is poured into pores of MSN in amorphous form. However, a weak and broad exothermic peak appears at 150 °C and 200 °C in MSN-FP, which indicates a noncrystalline state of drug entrapment into MSN. The 16.2% weight loss observed over the temperature range of 25–250 °C by TGA Figure 4 agrees with the calculated water content of 3.3% for the monohydrate. The melting peak depression of the MSN-FP may be attributed to the partial interaction of the drug with the surface of mesoporous nanoparticles which are rich in silica groups, causing partial amorphization of the FP drug [47]. 

### 3.4. Morphological Analysis

MSN prepared generally have high porosity. MSN showed a spherical shape after the particles were scanned using an electron microscope. The porous surface appeared on the MSN particle which was ensured by Nitrogen adsorption-desorption analysis, and it may be the result of phase evaporation during particle hardening. The controlled release properties of the drug particles that are entrapped in this MSN surface porosity make it a crucial component of the delivery system [48]. Figure 5B,C illustrated an electric microscopic view (SEM and TEM) of MSN-FP and the view showed well-dispersed MSN particles. API crystalline particles and commercial products of nasal spray also showed in Figure 5A which depicted large crystal particles within the formulation. 

### 3.5. X-ray Diffraction (XRD)

The XRD patterns for MSN, poloxamer, CMC, MSN-FP, and pure FP are presented in Figure 6. MSN showed low-intensity peaks in start at 2-theta values. Poloxamer and CMC also showed some low-intensity peaks in different values. Two broad peaks of MSN indicated its semi-order structure of mesoporous particles. 

FP exits sharp peaks with crystalline forms [43]. XRD patterns were similar to that of polymorphs with major peaks observed at characteristic diffraction peaks at 2-theta values of 9.3, 13.3, 15.8, 16.3, 20.9, 12.5, and 26.4 at 2-theta values [43], corresponding to the crystalline trend of FP. However, no prominent sharp peaks were noted except 13.3 and 16.1 in MSN-FP that confirmed the shifting of FP into amorphous form because of enlarged pore confinement with polymer matrix.

### 3.6. Nitrogen Adsorption-Desorption Analysis

Characterization of MSN-FP synthesized by the modified sol-gel method has high porosity (Kawashima et al., 1992). BET analysis was used for the evaluation of surface area (354.9 m^2^/g), pore size (6.1 nm), and volume of blank MSN. From the results, it was noted that significant reduction in BET surface area of MSN-FP than unloaded MSN due to the entrapped drug. Likewise, pore size (5.1 nm) and surface area (254.2 m^2^/g), were also decreased in MSN-FP as compared with unloaded MSN shown in Figure 7.

### 3.7. In Vitro Transcellular Permeability

This study was primarily aimed at verifying the in vitro transcellular permeability of formulated pH 7.4 MSN-FP through silica membranes. Degassed buffer was used to fill the clean, dried receptor cells and placed them in a magnetic block heated at 37 ± 0.5 °C for 15 min. A pre-hydrated silica membrane was used between the matching donor and acceptor compartments, an internal compartment filled with 1 g MSN-FP suspension in water. The opening was sealed with parafoil to prevent direct evaporation. A speed of 200 rpm was maintained to agitate the receptor compartment. A final sample with a volume of 0.5 mL was taken with a glass syringe for analysis at a wavelength of 254 nm using a UV spectrophotometer. A fresh preheated equal volume was reintroduced to maintain sample volume in the receiving chamber. In vitro permeation studies showed that the formulation could increase the time of FP administration compared to the pure drug suspension in water shown in Figure 8. These results suggest that MSN-FP has great potential for drug diffusion. 

### 3.8. In-Vitro Release Study

FP pure drug and formulation MSN-FP was synthesized with the aim to maintain the drug release for up to 8 h in order to deliver over-period action. The in-vitro FP release profile from the MSN-FP showed a biphasic type sustained release pattern (Figure 9). Pure drug release pattern is very slow in both pH (1.2 and 7.4) which means that FP is an insoluble drug (0.00402 mg/mL water solubility). In both media, it has shown almost the same pattern of poor solubility (30% release). Formulations (MSN-FP) exhibited a burst release pattern of the drug (35–40%) within the first hour of dissolution followed by the continual drug release pattern. A possible explanation for the initial burst release of MSN-FP is that some of the drug molecules may have adhered to the surface of the silica nanoparticles, causing a rapid increase in drug dissolution during the first hour of the release study. The negatively charged carboxyl group of FP interacts weakly with the negatively charged silanol group of silica nanoparticles which is unable to stop the burst release [49]. The later sustained release trend was attributed to the slow diffusion of the drug from silica nanoparticles up to 84–90% within both media. 

The above results suggested that the prepared silica nanoparticles could be applied as promising drug vehicles for sustained-release systems.

### 3.9. Kinetic Modeling

The release of FP from MSN-FP was investigated by using the different models on dissolution data; zero order, first order, Korsmeyer-Peppas and Higuchi. In the present study, MSN-FP was well suited for the zero-order release model (R^2^ = 0.9739) up to 8 h [50]. Thus, the rest of the mathematical model demonstrated the delivery of the drug incorporated inside the polymeric system after developing the pores through a diffusion mechanism [51]. Results Table 1 of R^2^ for Korsmeyer-Peppas (0.9184) indicated that MSN-FP followed the super case-2 transport mechanism too, controlled by using a matrix of water absorbency and subsequent relaxation [52]. 

### 3.10. Haemolysis Investigations

The compatibility of MSN-FP nano nasal spray with blood was investigated through hemolytic analyses. The results of hemolytic analyses (Figure 10) showed that MSN-FP nano nasal spray did not cause significant hemolysis after the administration at concentrations 100, 150, and 200 µg/mL, but the rate of hemolysis was noted towards high concentration i.e., 200 µg/mL. MSN-FP caused an immediate onset of hemolysis that soon reached a plateau of 27.25% hemolysis at 200 µg/mL probably due to its negative charge which might expel RBCs (−10 mV), which indicated that the extent of hemolysis was concentration-dependent (Figure 10). Results of the present study demonstrated that cell viability depended upon concentration and cells were alive after incubation with MSN-FP nano nasal spray which confirms blood compatibility. 

### 3.11. In Vitro Cell Viability Testing

According to the results of the MTT assay conducted on the human hepatoma cell line (HepG2) exposed to five different concentrations (100, 150, 200, and 400 µg/mL) of MSN-FP Nano nasal spray, the viability of cells was less at concentration ≥400 µg/mL of MSN-FP nano nasal spray formulation (Figure 11). The concentration of MSN-FP nano nasal spray in water is nearly nontoxic and biocompatible. Percentage cell viability was checked at 100, 150, 200, and 400 µg/mL of MSN-FP nano nasal spray and was found 91.28 ± 5.36, 82.42 ± 2, 71.13 ± 21, and 66.17 ± 8.97 µg/mL respectively demonstrating a dose-dependent pattern of cytotoxicity. The results of IC_50_ of the MSN-FP nano nasal spray after 24 h of exposure to HepG2 cells was 890.65 µg/mL that indicating a relatively non-toxic nature. 

### 3.12. Acute Toxicity Studies

In order to evaluate acute toxicity six healthy rats having an average weight of 150–170 g were chosen. These rats were divided into four groups with each group containing three rats. The first group was entitled as a control group while the second group was considered to be a diseased group in which diseased has been produced by of MSN-FP group. MSN-FP group rats were administered a nasally spray volume of nano nasal equivalent to 10 mg/kg, while the control group rats was given water and food. In order to evaluate different parameters like body weight; a physical examination of rats were done on the first day and 14th day of the study and all clinical findings were noted Table 2. About 1.0 mL of blood was drawn on the 14th day for biochemical and hematological analysis and the results of such analysis have been illustrated in Table 2. Both the control and MSN-FP nano nasal spray groups exhibit no morbidity or mortality. In comparison to the control group, no significant change in hemoglobin, Hct, RBCs, and WBCs parameters have been observed. Although there exist some minor differences between the two groups, however all hematologic parameters were within normal ranges.

### 3.13. Effect of FP-MG on IL-4, IL-5 mRNA Expression Level

Results indicate a significant increase in the mRNA level of IL-4 of group II (disease group) as compared to group I (control) (6.78 ± 0.127 vs. 5.38 ± 0.179). Pure drug-treated group III did not show a significant decrease in IL-4 levels as compared to group II (6.60 ± 0.277 vs. 6.78 ± 0.117) may be due to its delayed absorbance in rat lung tissues. However, there was a significant decrease in the IL-4 level of group IV which is MSN-FP as compared to group II (5.52 ± 0.159 vs. 6.878 ± 0.137) shown in Figure 12A. Similarly, the result showed in Figure 12B for the mRNA level of IL-5 of group II (disease group) as compared to group I (control) (5.82 ± 0.127 vs. 4.57 ± 0.179). Pure drug-treated group III did not show a significant decrease in IL-5 levels as compared to group II which is the disease group (5.61 ± 0.277 vs. 5.82 ± 0.127. However, there was a significant decrease in the IL-5 level of group IV which is MSN-FP as compared to group II which is a diseased group (4.79 ± 0.159 vs. 5.82 ± 0.127) shown in Figure 12B. The results showed MSN-FP when given intranasally to rats; inflammation significantly decreased total monocytes count, eosinophils and neutrophils in the blood as shown in Table 2. Inhibition of mRNA expression of IL-4 and IL-5 may be one of the mechanisms by which MSN-FP showed its anti-inflammatory effect. 

### 3.14. Physical Characterization 

After preparation and successful characterization of MSN-FP, it was converted into a nasal spray for use and was compared with commercial nasal spray for its physical post preparation characteristic. For nasal spray applications, MSN-FP nasal spray formulation was developed in liquid form. The pH of the nano nasal spray formulation was 6.1 (±0.1) according to USP, which is appropriate for nasal products. The viscosity of the formulation was 89 cpi at 25 °C. Results showed in Table 3. 

## 4. Conclusions

The present study suggested that MSN-FP Nano nasal spray is a well-optimized formulation for the potential treatment of severe Chronic Rhinitis. The in vitro studies demonstrated that the following formulation might sustain release for up to 8 h. The present study provides an innovative formulation in transdermal delivery of MSN-FP Nano nasal spray for the treatment of severe Chronic Rhinitis. Minor adaptations in various formulation factors can be fitted for other pharmaceutical usage and it will be a subject of our future studies.

## Figures and Tables

**Figure 1 jfb-13-00229-f001:**
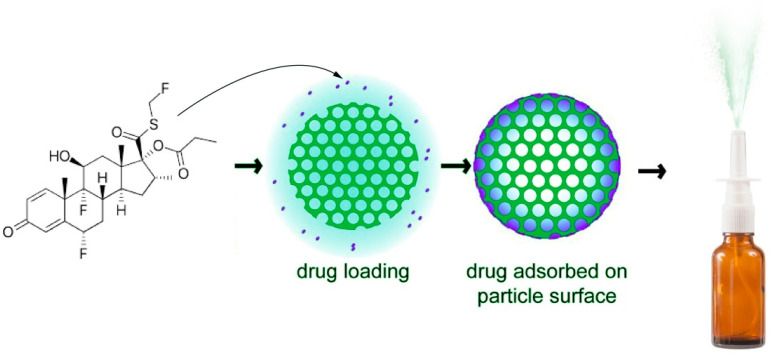
Schematic diagram of FP loading in MSN and converted into nasal spray.

**Figure 2 jfb-13-00229-f002:**
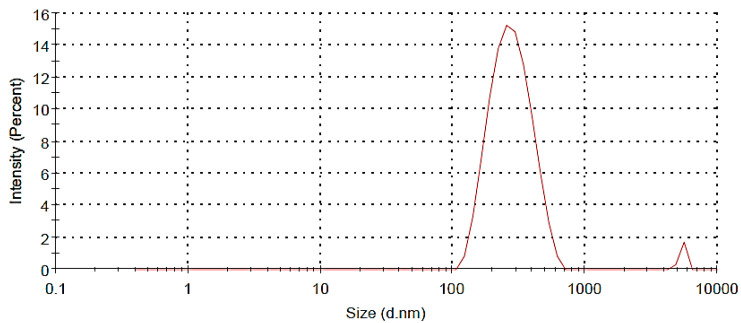
Size distributions of MSN-FP nano nasal spray.

**Figure 3 jfb-13-00229-f003:**
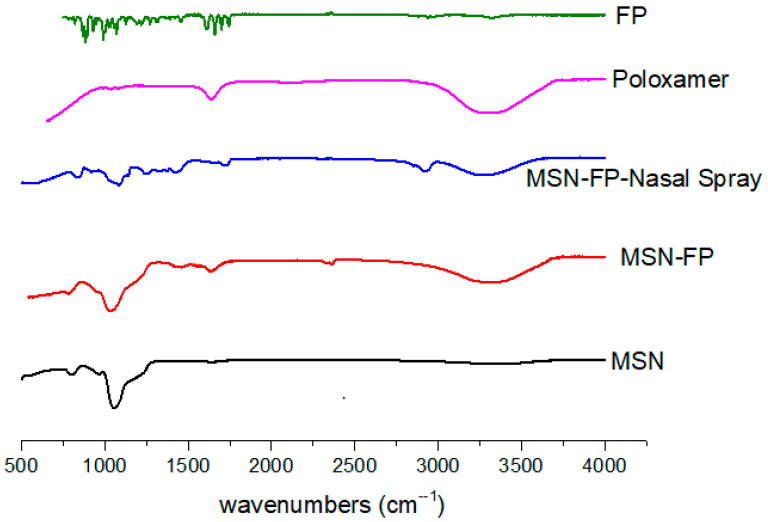
FTIR analysis of MSN, MSN-FP, MSN-FP-Nasal spray, poloxamer and FP.

**Figure 4 jfb-13-00229-f004:**
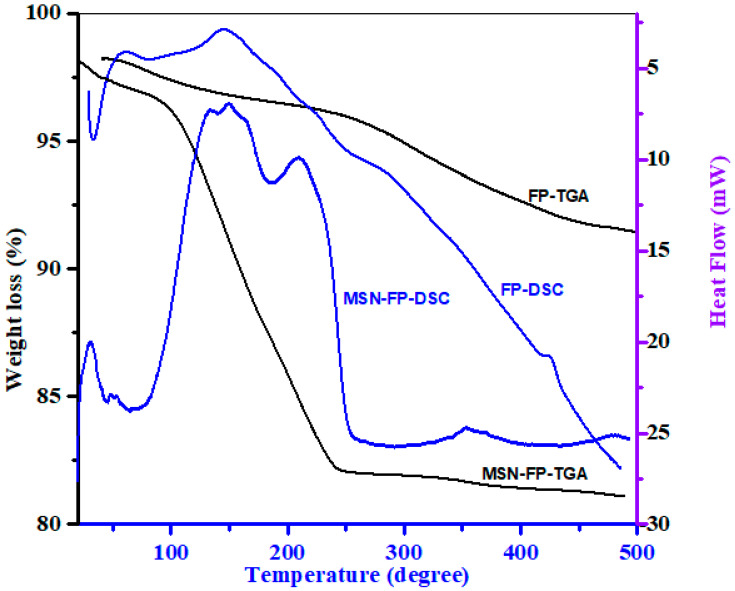
Thermal analysis of MSN, FP and MSN-FP.

**Figure 5 jfb-13-00229-f005:**
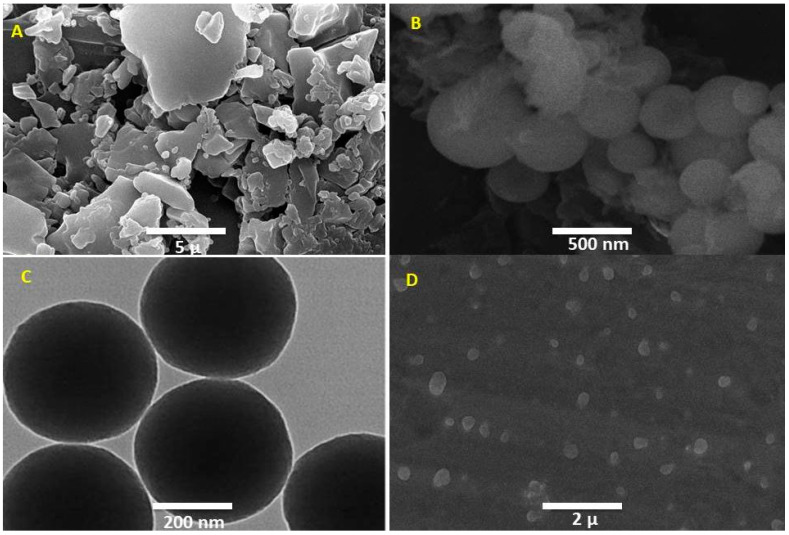
TEM and SEM images: (**A**) SEM image of API, (**B**) SEM Image of MSN-FP, (**C**) TEM images of MNS-FP and (**D**) SEM images of MSN-FP-Nasal spray.

**Figure 6 jfb-13-00229-f006:**
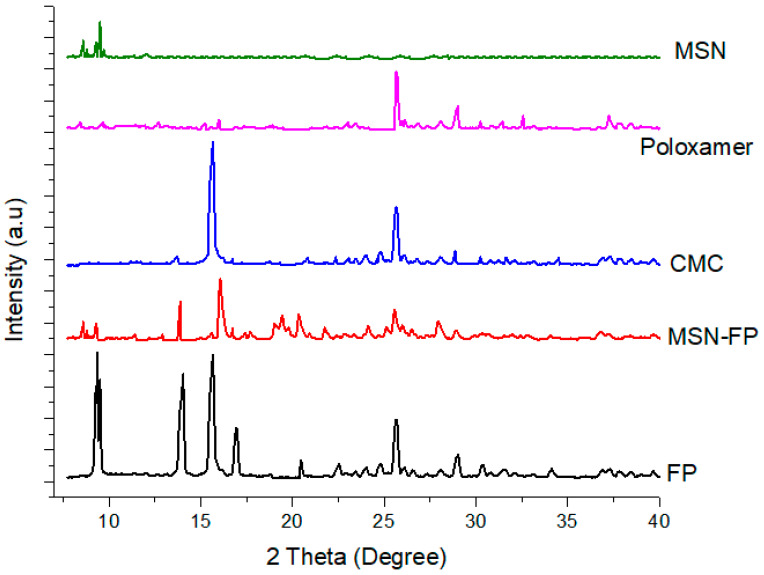
XRD patterns of MSN, poloxamer, CMC, MSN-FP and pure FP.

**Figure 7 jfb-13-00229-f007:**
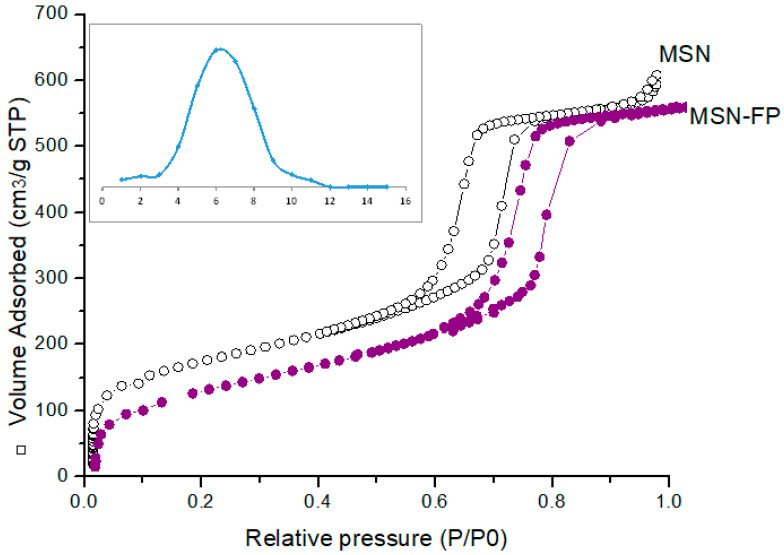
Nitrogen adsorption/desorption isotherms of MSN and MSN-FP.

**Figure 8 jfb-13-00229-f008:**
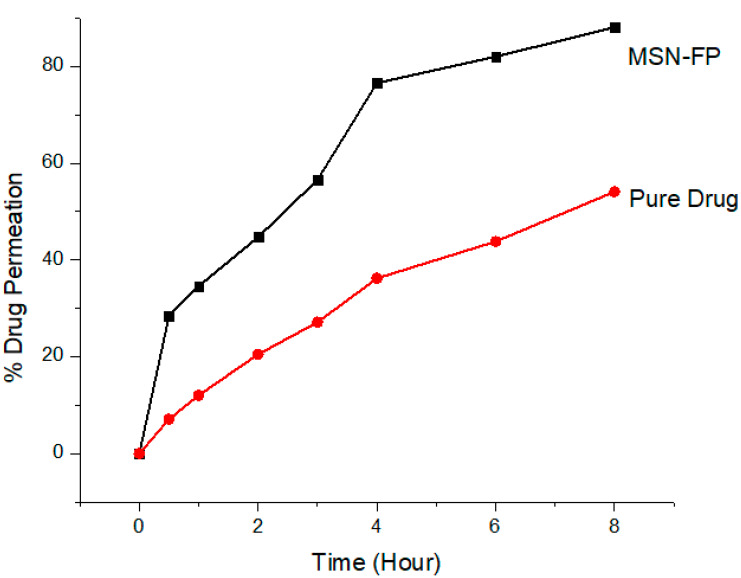
FP permeation from MSN-FP through silicon membrane compare with pure drug.

**Figure 9 jfb-13-00229-f009:**
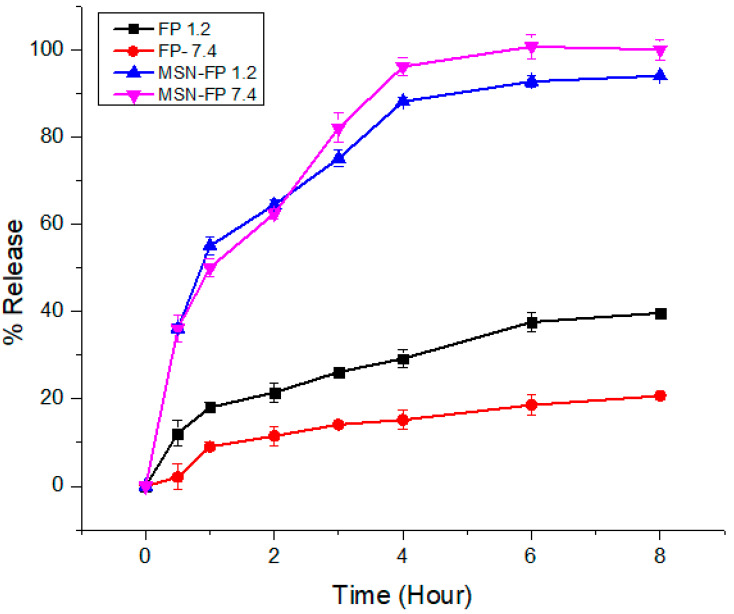
In-vitro drug release profile of pure drug and formulation studied in acidic buffer pH 1.2, and phosphate buffer pH 7.4 according to dialysis bag membrane method (n = 3).

**Figure 10 jfb-13-00229-f010:**
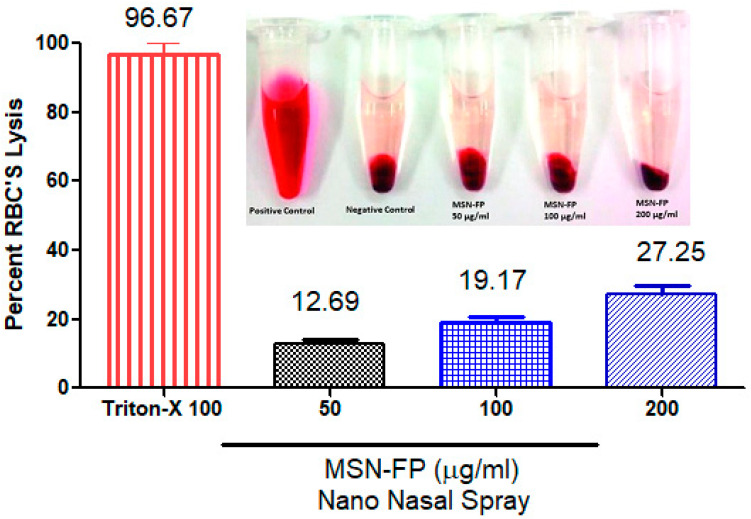
The compatibility of MSN-FP nano nasal spray with blood, Percentage toxicity induced at 50, 100 and 200 µg/mL of MSN-FP nano nasal spray.

**Figure 11 jfb-13-00229-f011:**
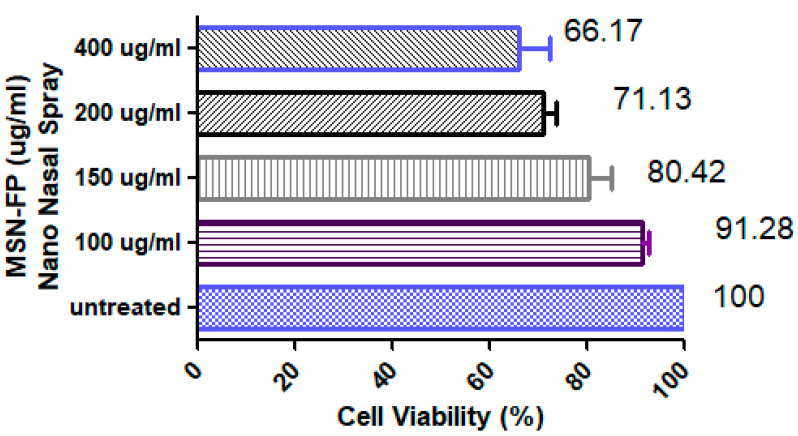
Cell Viability of MSN-FP nano nasal spray, Percentage induced at 50, 100, 200 µg/mL 400 µg/mL and of MSN-FP nano nasal spray.

**Figure 12 jfb-13-00229-f012:**
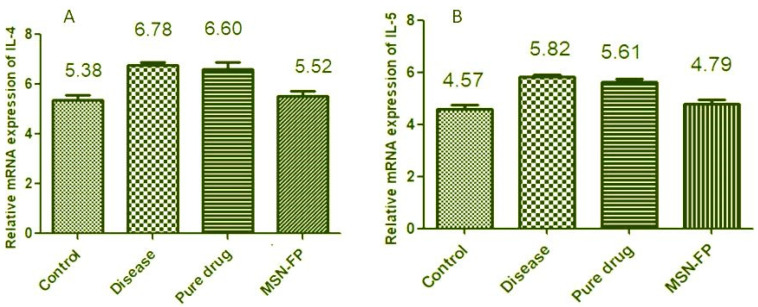
Effect of MSN-FP-Nasal-spray on IL-4 (**A**) and IL-5 (**B**) mRNA Expression Level.

**Table 1 jfb-13-00229-t001:** Kinetic Modeling of MSN-MP.

Models	Fitted Equation	R^2^
Zero Order	Q = 4.3963t + 4.7393	0.9739
First Order	Q = −0.1196t + 0.0704	0.8798
Higuchi	Q = 25.877t^1/2^ − 22.862	0.9135
Korsmeyer-Peppas	Q = 6.63902t^0.85^ − 2.83593	0.9184
Hixson Crowell	(1 − Q)^1/3^ = 0.0269t + 1.0006	0.9270

**Table 2 jfb-13-00229-t002:** Biochemical blood analysis.

Hematology	Group I	Group II	Group III	Group IV
	Control	Disease	Pure Drug	MSN-FP
WBCs × 10^9^/L	5.1 ± 0.41	6.8 ± 0.10	6.8 ± 0.17	5.3 ± 0.11
RBCs × 10^6^/mm^3^	5.96 ± 0.12	5.16± 0.19	5.56 ± 0.11	5.87 ± 0.12
Platelets × 10^9^/L	285 ± 0.31	269 ± 0.11	269 ± 0.21	272 ± 0.21
Monocytes (%)	7 ± 0.10	8 ± 0.21	8± 0.31	6 ± 0.13
Neutrophils (%)	42 ± 0.17	51 ± 0.41	51 ± 0.14	43 ± 0.21
Lymphocytes (%)	52 ± 0.91	63 ± 0.31	55 ± 0.21	51 ± 0.31
Eosinophils	3 ± 0.12	6 ± 0.18	2 ± 0.10	2 ± 0.17

**Table 3 jfb-13-00229-t003:** Physical parameters.

Formulation	Appearance	pH	Viscosity (cpi)
MSN-FP Nano-Nasal-Spray	Opaque	6.1 ± 0.1	89 ± 14
Commercial product (Ticovat)	Opaque	6.3 ± 0.2	77 ± 18

## Data Availability

Not applicable.

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
