# Peer review of "Developing of SiO2 Nanoshells Loaded with Fluticasone Propionate for Potential Nasal Drug Delivery: Determination of Pro-Inflammatory Cytokines through mRNA Expression"

_jfb, 2022, doi:10.3390/jfb13040229_

Round 1

Reviewer 1 Report

This manuscript by Mehmood et al reports using SiO2 nanoshells loaded with fluticasone Propionate for potential nasal drug delivery. The experiments were well designed and the data can support their conclusion. The authors also provide detailed experimental procedures so that others can readily repeat their results. The author also provided in-depth insight into the findings from this study. Therefore, the reviewer recommends the manuscript for publication after minor revision.

1.     In Figure 4, the colors of weight loss and heat flow should be switched.

2.     the author may need to provide the TEM image that can show the porous structure of MSNs.

3.     In line 23, it should be 400 ± 34 ??.

4.     In Figure 7, the X-axis should be 2 Theta (degree).

5.     In Figure 8, the value of X-axis should be from 0 to 1 if using relative pressure as the unut.

6.     In Figure 9, what does the “cumulative drug release per unit area” mean? The reviewer suggest the author change the unit to release efficiency.

7.     In Figure 12, it is suggested to add the original data points to the figure. Also, why doesn’t the triton X-100 treatment reach 100% of RBC lysis?

8.     In Figure 13 and Figure 14, it is suggested to add the original data points (viability) to these figures.

9.     In this manuscript, when the author mentioned “These properties enhanced drug loading and release capabilities. To have biomedical applications, such particles must have specific particle sizes, pore sizes, shapes, and pore geometries.”, the author may also need to cite the reference which can further support this claim: ACS Appl Mater Interfaces 2019, 11, 43835-43842. “Nanotechnology-based drug delivery can overcome certain anatomical, physiological, chemical, and clinical barriers associated with conventional dosage forms” the author may also need to cite the reference which can further support this claim. Ref: NMR in Biomedicine, 2013, 26, 1176-1185. Insoluble drugs require excipients to enhance solubility, such as surfactants, co-solvents, micellar solutions, chelating agents, and lipid formulations. Ref: Science Advances 2021, 7, abi9265.

Author Response

all figures has been set according to reviewer comments.

no more TEM images available, for pore size we used BET analysis 

all refernecs has been added as mentioned by reviewer

Graphs values has mentioned 

Reviewer 2 Report

The paper entitled " Developing of SIO2 Nanoshells Loaded with Fluticasone Propionate for Potential Nasal Drug Delivery: Determination of Pro-Inflammatory cytokines through mRNA Expression" describes the synthesis of mesoporous silica nanoparticles loaded with fluticasone propionate in order to be used for nasal drug delivery. I recommend the paper publication after major revisions and after responding the specified objections. I have the following comments and suggestions:

1. The clear statement of the novelty aspects of the study must be presented in Introduction studies. Why did the authors select these systems for drug delivery systems and what are their advantages compared  with other published systems in the field of nasal drug delivery.

2.  The manuscript has a number of grammatical and syntax errors. Please carefully read the manuscript and correct these errors. For example at line 187 the temperature for drug release is 37 °C but in manuscript is 37 0.5 °C. 

3. Lines 422-425:  I did not understand very well. MSN-FP nano nasal spray creates some haemolysis or not? In the first part of the sentence it is specified that MSN-FP nano system does not cause hemolysis, and in the second part it is stated that there is still a slight hemolysis ("The results of hemolytic analyses (Figure 11) showed that MSN-FP nano nasal spray did not cause haemolysis after the administration at concentrations 100, 150 and 200 mg mL−1 , but negligible haemolysis was noted even at high concentration i.e. 200 mg mL−1, blood cells exhibited well considerable tolerance to the complex".). Please check and correct. 

4. Immobilization of a drug on a carrier aims to remove the burst effect and to achieve a controlled or sustained release of the drug. In this study the release from the system obtained shows a rather large burst effect but also a high amount of drug than the free drug. In this case, what is the efficacy of the drug delivery system obtained?

5. The nasal mucosal pH is approximatively 5.5-6.5 and increases in rhinitis to 7.2-8.3. Why the release was made at pH=1.2, because this pH is specific to the stomach.

Author Response

1) he novelty aspects has been added in introduction along with advantages with references. Highlighted in red

2) Grammarly was used throughout manuscript for corrections. 37 0.5 was used for invitro study diffusion and dissolution, only 37 was used in cell study, however corrected in invitro where 0.5 was missing. 

3) Lines 422-425: was reviewed and corrected 

4) burst release was explained in paragraph

5) FP is insoluble on both ph thats why have checked its dissolution enhancement on both ph, high and low

Round 2

Reviewer 2 Report

The paper can be accepted in present form.